# The Relationship of Glutathione-*S*-Transferase and Multi-Drug Resistance-Related Protein 1 in Nitric Oxide (NO) Transport and Storage

**DOI:** 10.3390/molecules26195784

**Published:** 2021-09-24

**Authors:** Tiffany M. Russell, Mahan Gholam Azad, Des R. Richardson

**Affiliations:** 1Centre for Cancer Cell Biology and Drug Discovery, Griffith Institute for Drug Discovery, School of Environment and Science, Griffith University, Brisbane, QLD 4111, Australia; tiffany.russell@griffithuni.edu.au (T.M.R.); m.gholamazad@griffith.edu.au (M.G.A.); 2Department of Pathology and Biological Responses, Graduate School of Medicine, Nagoya University, Nagoya 4668550, Japan

**Keywords:** nitric oxide, dinitrosyl–dithiol iron complexes, free radicals, nitrogen monoxide, protein metal ion interactions, multi-drug resistance related protein 1, nitric oxide synthase, glutathione-*S*-transferase, vasodilation

## Abstract

Nitric oxide is a diatomic gas that has traditionally been viewed, particularly in the context of chemical fields, as a toxic, pungent gas that is the product of ammonia oxidation. However, nitric oxide has been associated with many biological roles including cell signaling, macrophage cytotoxicity, and vasodilation. More recently, a model for nitric oxide trafficking has been proposed where nitric oxide is regulated in the form of dinitrosyl-dithiol-iron-complexes, which are much less toxic and have a significantly greater half-life than free nitric oxide. Our laboratory has previously examined this hypothesis in tumor cells and has demonstrated that dinitrosyl-dithiol-iron-complexes are transported and stored by multi-drug resistance-related protein 1 and glutathione-*S*-transferase P1. A crystal structure of a dinitrosyl-dithiol-iron complex with glutathione-*S*-transferase P1 has been solved that demonstrates that a tyrosine residue in glutathione-*S*-transferase P1 is responsible for binding dinitrosyl-dithiol-iron-complexes. Considering the roles of nitric oxide in vasodilation and many other processes, a physiological model of nitric oxide transport and storage would be valuable in understanding nitric oxide physiology and pathophysiology.

## 1. General Biology of Nitric Oxide 

Nitric oxide (NO) is a diatomic gas that has traditionally been viewed as a toxic, pungent gas and is a product of ammonia oxidation [1]. In the past, it was believed that because poisonous agents such as carbon monoxide (CO) and cyanide (CN^−^) irreversibly bound to metal centers, that NO primarily functioned as a toxin [1]. However, the paradigm that NO existed purely as a toxic gas shifted in the 1980s, when it was discovered that NO was involved in key biological functions, such as signal transduction and cytotoxicity [2]. Since this time, NO has been associated with many other biological functions, including: neurotransmission [3,4], facilitation of the immune-response [5,6,7], blood pressure regulation [8,9], carcinogenesis [10,11], and macrophage cytotoxic effector functions [12,13,14].

NO exists as three different redox species, namely the nitrosonium cation (NO^+^), the nitroxyl anion (NO^−^), and the NO radical (NO^•^) [1] (Figure 1a). NO is highly reactive, allowing it to transition between redox species (oxidation to NO^+^ and reduction to NO^−^) and form NO_2_ after reacting with oxygen [1]. The NO^+^ serves as an effective oxidizing agent and is responsible for the *S*-nitrosylation of protein thiol groups [1]. It has been reported that NO^+^ has a shorter half-life in biological media compared to its redox counterparts, NO^−^ and NO^•^ [1].

Of note, NO^•^ avidly binds iron and is known to attack the active sites of a multitude of proteins, including heme and iron-sulfur clusters ([Fe-S]) [1]. It has been demonstrated that NO^•^ interacts as a ligand to form transition metal complexes with iron and glutathione (GSH), resulting in dinitrosyl–dithiol-iron complex (DNICs) formation, which is important for many physiological functions [15]. 

## 2. The Nitric Oxide Synthase Family of Enzymes

Endogenous production of NO occurs through the family of nitric oxide synthase (NOS) enzymes via the oxidation of l-arginine to l-citrulline (Figure 1b) [16]. There are three well-characterized NOS isoforms, namely neuronal ‘n’ NOS (or NOS I), inducible ‘i’NOS (or NOS II), and endothelial ‘e’NOS (or NOS III) [16]. Both nNOS and eNOS are commonly referred to as constitutive NOS (cNOS) and generate nanomolar concentrations of NO [17]. In contrast, activated macrophages expressing iNOS generate markedly and significantly larger concentrations of NO [18]. Each isoform of NOS utilizes l-arginine as the substrate for NO production and NAPDH and oxygen as co-substrates [17].

The NOS enzymes are all homodimers and facilitate electron transfer from nicotinamide-adenine-dinucleotide phosphate (NADPH) via flavin adenine dinucleotide (FAD) and flavin mononucleotide (FMN) to heme [17]. At the heme moiety, O_2_ is reduced and activated, and l-arginine is then oxidized to l-citrulline and NO [17]. The oxygenase domain containing the heme also binds the co-factor, (6R)-5,6,7,8-tetrahydrobiopterin (BH4), and l-arginine at sequences situated near the cysteine (Cys) ligand of the heme [16]. The activity of NOS enzymes is regulated by Ca^2+^ and calmodulin (CaM) [16,19,20]. When there is an increase in intracellular Ca^2+^ concentration, the affinity of CaM for eNOS increases, which then results in an influx of electrons from NADPH to the heme moiety of eNOS to initiate the conversion of l-arginine to l-citrulline [16,17].

The nNOS is expressed primarily in specific neurons within the central nervous system (CNS) and was the first NOS isoform identified in 1982 in neuroblastoma cells [21]. Since its discovery, this NOS isoform has been discovered in the spinal cord [22,23], certain epithelial cells [24,25], adrenal glands [26], peripheral nitrergic nerves [27], pancreatic islet cells [28], and vascular smooth muscle [29,30,31]. Furthermore, nNOS is largely associated with regulating physiological functions associated with memory, learning, and neurogenesis [32,33]. Animal studies reveal that the inhibition of NOS alters memory formation and synaptic plasticity, causing amnesia [32,33]. The expression of nNOS in the CNS has also been associated with blood pressure regulation [34,35]. Considering this, nNOS up-regulation in rats caused an increase in nitrergic vasodilation in superior mesenteric arteries in pre-hepatic portal hypertension [36]. Curiously, vascular smooth muscle cells that typically express eNOS also express low nNOS levels, which can partially regulate vasodilation in the absence of eNOS [37].

Inducible NOS is typically found at high levels in macrophages [16,18]. The expression of iNOS can be stimulated in cells or tissue by inflammatory cytokines (e.g., tumor necrosis factor-alpha (TNF-α); interleukin-1β (IL-1β), interferon-gamma (IFN-γ), etc.) and other agents such as bacterial lipopolysaccharide (LPS; Figure 2) [16,38]. However, iNOS can also be up-regulated by hypoxia [39], oxidative stress, and heat shock protein (Hsp70) [40]. The NF-κB and JAK-STAT signal transduction pathways are central to this regulation [41,42]. Thus, the inhibition of these pathways through agents like transforming growth factor-β1 (TGF-β1), glucocorticoids (GIC), glucosamine (GlcN), pyrrolidine dithiocarbamate (PDTC), and phosphatidylcholine-specific phospholipase C (PC-PLC) leads to a down-regulation or inhibition of iNOS expression (Figure 2) [43,44,45]. In addition, iNOS expression in cells can also be mediated by the MAPK pathway, which activates transcription factors such as activator protein-1 (AP1), activating transcription factor 2 (ATF2), and NF-κB [38,42].

Unlike nNOS and eNOS, iNOS is transcriptionally regulated, with CaM being permanently bound to iNOS [46]. Of note, iNOS induced in macrophages exposed to cytokines and/or LPS results in the generation of abundant NO [46]. As previously mentioned, NO has a high affinity for iron centers [1], which means that increased generation of NO via iNOS has the potential to attack and inhibit the iron-containing active sites of various proteins [1,47]. Examples of these proteins include (1) iron–sulfur cluster-dependent enzymes, such as those found in complexes I and II, which are crucial for mitochondrial electron transport; (2) ribonucleotide reductase, the rate-limiting enzyme of DNA synthesis; and (3) mitochondrial aconitase that plays a critical role in the Krebs cycle [18]. Studies have also indicated that enhanced NO generation in activated M1 macrophages disturbs tumor target cell DNA synthesis [48,49]. The inhibition of iron-containing enzymes and disruption of DNA synthesis contributes to the cytotoxic impact of NO on tumor cells [18].

Although eNOS was first identified in endothelial cells [16], it has since been identified in platelets [50,51], cardiac myocytes [52,53,54], neuronal cells [55,56,57], syncytio-trophoblasts (human placenta) [58], and kidney tubular epithelial cells [59]. The activity of eNOS is centrally modulated by Ca^2+^-activated CaM [60] (Figure 3). It has also been reported that the transmembrane protein, caveolin-1 (Cav-1), which is an integral component of caveolae [16], can be identified in a complex with eNOS that inhibits its activity (Figure 3). When eNOS is stimulated to increase intrinsic Ca^2+^, eNOS dissociates from Cav-1 and then binds CaM [60]. This effect is reversed when Ca^2+^ returns to its basal level (Figure 3). The inhibitory activity of Cav-1 is demonstrated by the increased endothelium-dependent relaxation of blood vessels of caveolin-1-deficient mice [61].

The regulation of eNOS can occur via several co- and post-translational modifications, including phosphorylation and protein–protein interactions (Figure 3) [60]. One of these proteins includes heat shock protein 90 (Hsp90), which is an allosteric modulator that assists in activating and (re)coupling eNOS [16]. Vascular endothelial growth factor (EGF), histamine and fluid shear stress results in the recruitment of HSP90 to eNOS, which promotes NO production [62]. Furthermore, García-Cardeña and colleagues have identified that vascular endothelial growth factor, G-protein and mechano-transduction pathways are responsible for promoting the activity of Hsp90-mediated eNOS activation [62].

While the NOS enzymes are the major source of NO, more recently in vitro and in vivo studies have revealed that nitrite and nitrates can be metabolized directly to NO via xanthine oxidoreductase [63,64]. Of interest, xanthine oxidoreductase is a homodimer of approximately 300 kDa, with each subunit containing four redox centers, namely a molybdenum cofactor, one FAD, and two Fe_2_S_2_ clusters [63]. The latter two clusters could potentially be involved in feedback regulation mediated through the ability of NO to attack these moieties, which is discussed in greater detail below in Section 3.2.

## 3. Physiological Roles of NO

The biochemistry of NO is complex in cells but can be divided into two major reactions that result in *S*-nitrosylation or its direct ligation with iron forming DNICs. Both of these reactions are discussed in detail below.

### 3.1. NO and S-Nitrosylation

NO can induce *S*-nitrosylation of proteins [65,66,67] and low molecular weight thiols such as GSH, leading to *S*-glutathionylation [68]. Under physiological conditions, cells typically function in a reduced environment to maintain low levels of reactive oxygen species (ROS) and reactive nitrogen species (RNS), which are toxic to cells [68]. Thus, a range of cellular anti-oxidants antagonize the oxidative activity of ROS/RNS and include superoxide dismutase (SOD), vitamins E, A, and C, and the major thiol, GSH [68].

*S*-nitrosylation is an enzyme-independent modification involving the reversible binding of NO to a sulfur atom to create an *S*-nitrosothiol, which can function in other roles; for example, in the transcriptional modulation of hypoxia-inducible factor-1α (HIF-1α) through the *S*-nitrosylation of its free thiol groups [69,70]. This reaction relies on the redox micro-environment and is regulated by GSH/GSSG, NADPH/NADP+, etc. [68]. Another example of the regulatory activity of *S*-nitrosylation is demonstrated by the fact that the apoptotic pathway effector, pro-caspase-3, is *S*-nitrosylated at the active site Cys, which inhibits the enzyme’s activity and prevents apoptosis [71]. The intracellular reactions of NO to generate *S*-nitrosothiols or DNICs do not occur in isolation as RSNO formation can be influenced by DNIC synthesis [72]. Clearly, RSNO generation can also be independent of DNICs [73].

It is well known that GSH is the primary thiol in the cytoplasm that functions to maintain the reduced micro-environment of the cell [74]. It usually exists in its reduced form (GSH), but can exist in its oxidized state, GSSG, that is generated under multiple conditions, including via glutathione peroxidase (GPxs) or interactions with ROS/RNS [75]. *S*-nitrosylated proteins can be reduced by GSH, and this reaction results in the formation of *S*-nitrosoglutathione (GSNO) [68]. Furthermore, it has been reported that GSNO can serve as an intermediate for the synthesis and degradation of *S*-nitrosothiols and can transfer NO to thiols, or thiol-containing proteins via transnitrosylation [68].

### 3.2. NO and Its Interaction with Cellular Iron

A key characteristic of NO is its ability to bind iron centers in proteins with a high affinity [76]. This effect of NO is central to many of NO’s physiological functions, particularly servo-regulation and cytotoxicity [14,76,77,78,79,80,81]. There are three main types of NO–iron reactions: (1) direct ligation of NO with the iron center; (2) oxidation of iron with the consumption of NO; and (3) the reduction of metals to limit the formation of oxidants and oxidative stress [78]. NO can react avidly with iron to generate stable iron nitrosyl complexes, which includes NO’s reaction with proteins containing heme moieties [78], or iron stored in molecules such as ferritin [82]. The reaction of NO with heme moieties, such as those in soluble guanylate cyclase (sGC), cytochrome P450, and NOS are key to the many physiological roles of NO.

NO modulates many signaling pathways and processes through binding to the heme prosthetic group of sGC [14,76]. This reaction produces an iron-nitrosyl complex that leads to the activation of the enzyme that then generates cyclic guanosine monophosphate (cGMP) [83,84,85,86,87,88]. The cGMP is a secondary messenger responsible for regulating several downstream processes, including vasodilation, retinal phototransduction, calcium homeostasis, and neurotransmission [83,84,85,86,87,88]. The concentration of NO needed to activate cGMP is relatively low (EC_50_ = 100 nM), highlighting the impact of NO produced from cNOS, which generates very low NO levels [78].

Another key NO-heme association is the reaction between NO and cytochrome P450 (P450). This protein belongs to a group of enzymes involved in the metabolism of exogenous and endogenous compounds [89], particularly certain drugs and xenobiotics [90]. Unlike the interaction with sCG, NO inhibits P450 activity via two different mechanisms, reversible and irreversible inhibition [91]. For reversible inhibition, NO binds to the heme of P450 to inhibit oxygen-binding, and thus, catalytic activity [91]. However, irreversible inhibition occurs via reactive nitrogen oxide species (RNOS), which have been proposed to remove heme from the enzyme [91]. It has been postulated that NO binds to the heme of P450, which shifts the Cys ligand from the iron, allowing it to be oxidized, preventing its re-binding to the heme iron [78].

Cellular iron homeostasis is controlled by various feedback systems that regulate the expression of proteins that monitor iron uptake, release, and storage. Two key proteins in iron homeostasis are iron regulatory protein 1 (IRP1) and iron regulatory protein 2 (IRP2), which bind iron regulatory element (IRE) sequences in 3′- and 5′-untranslated regions (UTRs) of mRNAs of proteins involved in iron metabolism (Figure 4) [92]. The RNA-binding activity of IRP1 and IRP2 are regulated very differently. In the case of IRP1, its RNA-binding activity is ablated by the formation of a [4Fe-4S] cluster in the presence of high cellular iron, while for IRP2 it is degraded by the proteasome under this condition [14].

In terms of the effect of the IRPs on IRE-containing mRNA’s, under conditions of cellular iron deficiency, IRP-binding to the IREs in the 3ꞌ-UTR increases *transferrin receptor 1* (TfR1) mRNA stability, leading to increased uptake of iron from transferrin (Tf) via receptor-mediated endocytosis of the Tf-TfR1 complex [93,94,95]. On the other hand, low cellular iron levels lead to the binding of the IRPs to the IRE in the 5ꞌ-UTR of *ferritin-H* and *ferritin-L* mRNA, resulting in the inhibition of translation [96]. A schematic explaining the mechanism of this interaction between IRP1 and IREs is summarized in Figure 4.

Furthermore, apart from *TfR1* and ferritin mRNAs, there are multiple other mRNAs regulated by the IRP-IRE mechanism, including *ferroportin*, the IRE-containing splice variant of the *divalent metal transporter 1* (*DMT1*) [97,98], *hypoxia-inducible factor 2α* (*HIF2α*) [99] amongst others. Most recently, a new 5′-IRE containing mRNA has been identified, namely that encoding tetraspanin, CD63, which regulates exosome formation and the secretion of iron-loaded ferritin [100]. Of note, studies have shown that NO induces the RNA-binding activity of IRP1 and IRP2 [76,101,102,103].

Regarding the mechanism of this latter response, as demonstrated in Figure 5, in the presence of a [4Fe-4S] cluster, IRP1 cannot bind IRE [104]. However, when iron depletion occurs, the [4Fe-4S] cluster is lost, enabling IRP1 to bind to the IRE motif [104]. In the presence of NO, two mechanisms result in the loss of the [4Fe-4S] cluster, namely: (1) direct attack of NO inducing disassembly of the cluster; and (2) an indirect mechanism where NO binds intracellular iron to induce its release from the cell [103,105,106]. This results in iron depletion that prevents biogenesis of the [4Fe-4S] cluster (Figure 5) [103,105,106].

On the other hand, IRP2 is degraded by a proteasomal mechanism when iron is present, and thus can bind IREs under conditions of iron deficiency [104]. For IRP2, it has been discovered that NO and peroxynitrite (ONOO^−^) decrease IRP2–RNA binding activity [107,108,109], which may be induced by NO^+^-mediated degradation [109]. Although NO’s effect on IRP-RNA-binding activity is well characterized, the biochemistry of NO is complex, as it has multiple iron-containing targets within cells. Richardson and colleagues identified a minimal correlation between increased TfR1 expression and Tf-mediated iron uptake [102], which could be due to NO inhibiting ATP synthesis, causing a disruption in ATP-dependent iron uptake from Tf [110,111].

Taking into account NO’s functional effects on the direct interaction with cellular iron, Hibbs and colleagues proposed a molecular mechanism for the cytotoxicity of activated macrophages against tumor cells that involves targeting critical iron-containing proteins [13]. In this case, when activated macrophages were co-cultivated with iron-59-labeled tumor cells, there was a marked loss of intracellular iron content within a 24 h incubation [13]. This decrease in cellular iron was associated with the inhibition of tumor cell DNA synthesis and mitochondrial respiration [13]. It has been suggested that the iron released from tumor cells, mediated by NO-activated macrophages, originated from [Fe-S] containing proteins from the electron transport chain [112,113]. This reaction leads to NO and iron forming complexes with GSH or Cys to result in DNICs [112,114,115]. Under NO stress, protein-bound DNICs have been observed in prokaryotic (e.g., *Escherichia coli*) and eukaryotic cells, with [Fe-S] proteins being a major target of NO [116].

Another important consequence of NO’s ability to bind intracellular iron is the up-regulation of N-myc downstream-regulated gene-1 (NDRG1), which suppresses tumor cell signaling and migration [117,118]. NDRG1 is a ubiquitous cellular protein that is tightly regulated by N-myc [119], histone acetylation [120], intracellular calcium [121,122], and cellular iron levels [123,124]. The up-regulation of NDRG1 is observed in neoplastic cells in response to iron chelators that induce its expression via HIF-1α [124,125,126]. The activity of NDRG1 suppresses the oncogenic activity of a multitude of signaling pathways associated with cancer metastasis and tumorigenesis, including NF-κB [127,128], P13K/AKT/mTOR [126,129], Wnt signaling [130], and Ras/Raf/MEK/ERK [126,131] pathways.

It has been demonstrated that NDRG1 up-regulation after exposure of cells to NO is the result of it depleting the intracellular labile iron pool via inducing iron mobilization from cells [103,132,133,134]. Hickok and colleagues identified that interactions between NO and the iron pool and the subsequent generation of DNICs were ‘key determinants’ in NDRG1 up-regulation in cultured breast cancer cells [117]. It has been speculated that the relationship between NDRG1 and NO could contribute to the anti-neoplastic effects of activated cytotoxic macrophages [135].

Contemplating the marked interaction of NO with iron described above, it is possible that formation of DNICs in cells could affect the utilization and trafficking of other metal ions such as copper, which is closely linked with the metabolism of iron [14]. 

## 4. Biological Functions of DNICs

As eluded to above, DNICs are complexes composed of NO, iron, and other ligands (Figure 6) found in cells or tissues after exposure to NO, or NO-generating agents [115,136,137,138,139] and can also be observed physiologically [137]. These complexes can be directly identified via their fingerprint signal upon examination with electron paramagnetic resonance (EPR) spectroscopy (*g* = 2.03), which is the only technique that can specifically identify DNICs in living cells [140].

Studies using EPR indicate an equilibrium between free DNIC complexes and protein-bound DNICs [15,141,142,143]. Importantly, DNICs, like RSNOs, are endogenous NO carriers and are formed via the reaction of iron and NO with low molecular weight thiols, peptides [144,145,146], and some proteins [47,116,147]. The binding of DNICs to proteins is facilitated by substituting a thiol ligand from the “free” DNIC with Ser, Tyr, or Cys [140]. However, dinitrosyl-diglutathionyl iron complexes (DNDGIC) are generally considered as the most abundant DNIC within cells [148], and this can be attributed to the high concentration of GSH in cells (2–10 mM) [149].

Low molecular weight DNICs with thiol-containing ligands exist as mononuclear (EPR-active; M-DNIC) and binuclear (diamagnetic; B-DNIC) complexes [150,151,152,153]. The transition from M-DNIC to B-DNIC occurs when there is a decrease in intracellular free thiols [151]. When the concentration of free thiols, particularly thiols ionized at the sulfur atom, increases, B-DNICs are converted to the M-DNIC form [151]. Both complexes contain iron-dinitrosyl fragments as active components, which Roussin and colleagues identified in synthetic salts [154]. The characteristic release of NO from these complexes, and their ability to bind heme groups of regulatory proteins such as sGC, defines a key component of the biological activity of DNICs [155].

Studies have indicated that DNICs represent a “working form” of NO, which is non-toxic and functions as a stable currency of NO and a storage mechanism [156,157,158,159,160,161,162]. In this form, the half-life of NO is dramatically increased from seconds to hours [163], and these complexes can modulate several regulatory activities by releasing NO [164]. Hickok and colleagues have previously demonstrated that DNIC formation can be impacted directly by altering O_2_ levels [163]. Due to the extended half-life of NO in DNICs, DNICs may function to capture NO and transfer it to thiols for *S*-nitrosothiol production [163]. It has been further suggested that DNICs may function as NO generators to bolster NO activity [163]. This proposal is supported by the involvement of DNICs in sGC activation and their ability to induce vasorelaxation [165,166,167,168,169].

With respect to the effect of DNICs on cellular redox status, free DNICs have been shown to disrupt the cellular redox environment by irreversibly inhibiting glutathione reductase [140,156], which could lead to a more oxidized micro-environment. In contrast, the possible antioxidant functions of DNICs were demonstrated by their ability to inhibit Cu^+2^-induced peroxidation of low-density lipoprotein (LDL) and prevent ROS generation during co-oxidation of lecithin-containing liposomes and glucose [170]. Another antioxidant mechanism was proposed by Shumaev et al. [171], who demonstrated that DNICs reduce oxoferryl-myoglobin more efficiently than *S*-nitrosoglutathione and GSH. It was also identified that DNICs suppress the thiyl radical of GSH in medium containing metmyoglobin and t-butyl hydroperoxide [171].

The potential ability of DNICs to influence redox chemistry suggests that they could possess cytotoxic activity in cells unless they are carefully regulated. In fact, DNICs induced apoptosis in Jurkat cells in a concentration- and time-dependent manner, despite overexpression of the anti-apoptotic molecule, Bcl-2 [172]. On the other hand, the presence of free DNIC complexes was found to have no pro-apoptotic effect on HeLa cells, while GSNO induced pro-apoptotic activity [173]. These conflicting studies demonstrate that DNICs may interact in different ways depending on the cell-type.

Another study conducted by Konstantin and colleagues showed that exogenous chelators induced DNIC degradation, which caused NO release and apoptosis [174]. This later observation suggests that exogenous metal-binding ligands and those found endogenously, such as transferrin and lactoferrin, may influence DNIC stability. Furthermore, since iron(II), *S*-containing ligands (e.g., GSH, cys, etc.), and NO react to generate DNICs, any agent that alters this equilibrium could influence DNIC stability. As such, excess Fe, or even molecules that directly bind DNICs, such as glutathione-*S*-transferases (GSTs) [140], could lead to alterations in NO bioactivity.

Since DNICs form the majority of NO in cells [163], and because of their roles as an NO currency, a model for NO storage and transport in the form of DNICs has been proposed [132,135,175] (Figure 7). The studies that have led to the generation of this model are described in the sections below.

## 5. GSH and Energy Metabolism Are Essential for NO-Induced Iron Efflux from Cells

The development of the model in Figure 7 originally came from biochemical studies trying to dissect the mechanism responsible for energy-dependent iron mobilization from cells mediated by NO [133,176]. It was demonstrated that while d-glucose could be metabolized and stimulate NO-mediated iron release, monosaccharides and disaccharides that could not be metabolized or transported into cells had no impact [133]. This latter study also demonstrated that NO-mediated iron efflux was greatly reduced after incubating cells with cytochalasin B, a glucose transport inhibitor [133].

There are two key pathways responsible for the metabolism of d-glucose: glycolysis/the tricarboxylic acid (TCA) cycle and the pentose phosphate pathway (PPP), the second of which produces NADPH, which is essential for redox metabolism [175]. The metabolism of d-glucose by either pathway has the potential to influence NO-mediated iron metabolism [133,135]. However, incubation of cells with the TCA cycle intermediates, citrate or pyruvate, did not affect NO-mediated release of iron from cells [133]. On the other hand, pre-incubation with L-buthionine-[S,R]-sulfoximine or diethyl maleate, which deplete GSH levels, suppressed NO-mediated iron release from cells [133].

Furthermore, treatment with agents that increase cellular GSH levels, such as *N*-acetyl-l-cysteine (a GSH precursor), reversed the impact of the GSH-depleting agents, resulting in increased NO-mediated iron mobilization from cells, indicating that GSH is required for this process [133]. Of interest, NO-mediated GSH-depletion results in subsequent activation of the PPP, probably as a rescue response to reconstitute GSH levels [180].

## 6. Multi-Drug Resistance-Related Protein 1 (MRP1) Mediates the Release of Iron and GSH from Cells as DNICs 

As described above, GSH and metabolic energy are required for NO-mediated iron efflux from cells [133]. Research from our laboratory proposed that DNICs composed of GSH, iron and NO are released from cells in an energy-dependent manner [133]. Taking this into account and the fact that GSH complexes of arsenic and antimony are transported by MRP1, studies examined MRP1’s role in DNIC transport [132,134,135,162,175]. These experiments demonstrated that the release of iron and GSH from tumor cells was mediated by MRP1, which is a prominent ATP-dependent GSH exporter [132,134,135,162,175]. Of note, MRP1 is a well-known drug efflux pump, which couples ATP hydrolysis to the export from cells of anti-cancer drugs that are conjugated to GSH [149,181,182] (Figure 8).

The role of MRP1 in NO-mediated iron release from cells was first explored in expression studies using MCF7-VP human breast cancer cells hyper-expressing MRP1 relative to MCF7-WT (basal expression of MRP1) [134,183]. In this study, MCF7-VP cells overexpressing MRP1 demonstrated a 3- to 4-fold increase in NO-mediated iron and GSH release compared to the wildtype (MCF7-WT) cells within a 4 h incubation [134,183]. It was found that NO-mediated iron and GSH release was significantly inhibited in the presence of the GSH-depleting agent, BSO, and MRP1 transport inhibitors, such as MK571, probenecid, and difloxacin [134,183].

Furthermore, through EPR spectroscopy that detects DNICs via their “fingerprint” spectral signature, it was identified that cells incubated with MRP1 inhibitors demonstrated a more intense DNIC signal when exposed to NO [134,183]. These data indicated that inhibition of MRP1 transport activity which resulted in the suppression of a DNIC export, resulting in higher intracellular DNIC levels, leading to a more intense EPR signal. These results were later confirmed in a study examining the inhibition of NO-mediated iron release from murine embryonic fibroblasts derived from MRP1-knockout mice [175]. Hence, MRP1 was key to the NO-mediated iron and GSH export from cells.

More recently, considering the important role of NO in the cytotoxic response of macrophages against tumor cells, studies explored whether MRP1 could be involved in NO transport [132]. These studies demonstrated that MRP1 expression was markedly increased after macrophage activation and that NO-mediated iron release was inhibited by *Mrp1* silencing or the MRP1 inhibitor, MK571 [132]. This latter result suggested that MRP1 was responsible for DNIC efflux. Furthermore, DNIC accumulation within macrophages occurred when *Mrp1* was silenced [132], which is consistent with the inhibited transport of the DNIC out of cells, further supporting MRP1’s role in DNIC efflux, and thus, iron and GSH release from cells.

As noted above, NO-mediated iron and GSH export is facilitated by MRP1 expression in multiple cell-types [134,183]. This result is significant because DNICs have a markedly increased half-life relative to “free” NO [184], and the transport of NO as a DNICs could modulate many of NO’s biological functions, such as vasodilation. Given NO’s role in this latter process [185,186], eNOS generated NO and DNICs within endothelial cells, and their respective export by MRP1 into smooth muscle cells can be hypothesized to result in smooth muscle relaxation [169].

It has been proposed that GSH, in addition to forming DNICs with iron and NO for export by MRP1, may be required for cellular iron release from proteins that interact with NO, such as [Fe–S] proteins [187,188]. In fact, it has been proposed that there is an intracellular equilibrium between protein-bound and low molecular weight DNICs [177] and that GSH converts protein-bound DNICs to low molecular weight DNICs to facilitate their export by MRP1 [134,183].

## 7. MRP1 Forms an Integrated Detoxification System with GSTs in Drug Resistance

MRP1 is a major transporter of GSH and GSH conjugates [189]. Since the discovery of MRP1, its transport roles have been associated with detoxification mechanisms against xenobiotics, endobiotics [190], and oxidative stress [191]. Furthermore, MRP1 is established as a central protein for multiple drug resistance in cancer [190,192,193,194], with studies indicating a strong correlation between MRP1 hyperexpression in tumors and poor prognosis [195,196]. MRP1 has been shown to work together with GST enzymes to form multi-drug resistance to a variety of anti-cancer drugs, including paclitaxel [191], vincristine [197], etoposide [197,198], cisplatin [199], mitoxantrone [200], chlorambucil [201] and melphalan [202].

GSTs are a superfamily of multi-gene detoxification isoenzymes that metabolize xenobiotic and endobiotic molecules in cells via their ability to conjugate GSH to these compounds [203]. The GSTs are expressed in most living organisms and represent 1% of cellular protein [204]. The different members of these classes of GSTs are distinguished by their sequence similarity and immunological cross-reactivity [205,206].

Human cytosolic GSTs have been divided into seven classes (α, µ, π, σ, θ, ω, and ξ) [207]. As illustrated in Figure 8, MRP1 and GST enzymes have been primarily associated with detoxification mechanisms, in which GST enzymes conjugate GSH to toxins to target them for export out of the cell by MRP1 [201,208,209]. Reflecting upon the synergistic roles of GSTs and MRP1 in detoxification and metabolism [201,208,209] and that MRP1 transports iron and GSH in a form consistent with DNICs [134,175], it is significant that the most abundant GSTs, α (GST A1-1), µ (GST M1-1), and π (GST P1-1), bind DNDGICs with marked affinity (*K_d_* 10^−9^ to 10^−10^ M) [160,210].

## 8. The Potential Intermediary or Storage Role of DNICs by GST Enzymes

The role of GSTs in detoxification may be significant in NO metabolism, given that the binding of with DNICs directly to GSTs results in a markedly increased half-life (4.5–8 h) [210] and a significantly increased half-life than “free NO” (i.e., 2 ms to 2 s) [211]. A crystal structure of GST P1-1 complexed with a DNIC at high resolution has been revealed by Cesareo and colleagues [210]. In this complex, it was identified that the active site residue, Tyr-7, coordinates to the iron atom of the DNICs by substituting one of the GSH ligands [210]. This latter study demonstrated that site-directed mutagenesis of this latter Tyr residue with other residues (i.e., His, Cys, or Phe) resulted in a significant decrease in DNIC-binding activity [210]. Furthermore, the half-life of the complex within human placenta and rat liver homogenates revealed that GST-P1-1 stabilized the DNICs for 8 h, while GST-A and GSTM stabilized DNICs for 4.5 h [184].

The binding of DNICs to GST A within hepatocytes has also been explored [140,212]. It was proposed that DNIC complexes were formed in response to NO-induced iron chelation from ferritin and Tf [140]. It has been suggested in cells that NO intercepts iron after it has been released from Tf in endosomes [111] and binds to iron liberated as a result of ferritin degradation in lysosomes [135,213]. This observation is consistent with a study conducted by Lewandowska and colleagues, indicating that DNIC formation was the result of metalloprotein degradation in lysosomes [214]. Consistent with this latter conclusion, it was also demonstrated that the lysosomal acidification inhibitor, NH_4_Cl, prevented DNIC formation [214].

Complete understanding of the functional roles of DNICs bound by GST enzymes remains in its infancy, although some studies have indicated that GSTs may function as storage enzymes or as protective mechanisms within cells [140,175,184]. A 2007 study identified that although DNICs irreversibly inhibit glutathione reductase, the activity of these enzymes could be stabilized in the presence of GSTA [140]. However, when the concentration of DNICs exceeded the binding capacity of the GST enzymes, inactivation of glutathione reductase occurred [140]. A later study revealed that 20% of these DNIC-GST complexes are associated with subcellular components, notably the nucleus, which may suggest a protective mechanism against DNA damage [212].

Our laboratory has proposed an integrated model in which GSTs function as a storage protein for DNICs within the cell that are subsequently transported out by MRP1 [175]. In fact, it was demonstrated that GST P1-1 expression significantly decreased NO-induced iron release from tumor cells via MRP1 [175]. In this study, experiments were conducted with either (1) MRP1 hyper-expressing MCF7-VP cells; or (2) the relevant control cell-type (i.e., MCF7-WT cells without MRP1), with both cell-types being transfected to overexpress GST A1-1, M1-1, or P1-1 [175]. It was identified that MCF7-VP cells overexpressing GST P1-1 and treated with GSNO significantly decreased NO-mediated iron efflux compared to cells overexpressing GST M1-1 and A1-1 [175]. This result suggested that only GST P1-1 was capable of binding DNICs intracellularly in tumor cells, which prevented their release by MCF7-VP cells hyper-expressing MRP1. Notably, these results were also replicated in the same cell-type transfected with NO-generating iNOS, which is more physiologically relevant [175].

Furthermore, EPR and fast-pressure liquid chromatography were used to verify that this decrease in NO-mediated iron release by MRP1 in GST P1-1 over-expressing MCF7-VP cells was due to an increase in DNIC-GST P1-1 complex formation [175]. The GST P1-1-DNIC complexes may serve as a ‘storage sink’ for NO within the cell to stabilize NO for various cellular functions [132,135].

It was also demonstrated in MCF7-VP cells overexpressing MRP1 that iron-59 accumulated within GST P1-1-containing fractions [175]. These data strongly supported the hypothesis that DNIC-GST P1-1 binding inhibits NO-mediated iron release. Furthermore, EPR studies revealed that the MCF7-VP cells transfected with GST P1-1 demonstrated a significantly greater DNIC signal, suggesting binding of DNICs to this protein [175]. These results were supported by X-ray crystallography studies conducted by Cesareo et al. [210], that demonstrated DNIC-GST P1-1 complexes in the isolated protein, but not within cells. These investigations collectively demonstrate that GST P1-1 sequesters NO as DNICs in cancer cells, preventing DNIC export out of the cell by MRP1 [175,210].

Studies examining rat hepatocytes and liver homogenates also identified DNIC complexes bound to GST A [140]. Implementing EPR spectroscopy and enzyme activity measurements the binding of DNDGIC) bound to GST A was found to exhibit an extraordinary high affinity (*K_d_* 10^−10^ M) [140]. Taking into account that GST enzymes bind DNICs with slightly differing affinities (*K_d_* 10^−9^ to 10^−10^ M) [160,210] and that GST A is highly expressed in hepatocytes (0.3 mM concentration) [140], it may be that different GST enzymes are necessary for protection against NO cytotoxicity within different cell types. Further investigations using a variety of physiologically relevant cell-types that express multiple GST isoforms are required to understand how GST enzymes interact with DNICs and MRP1.

Taking into account the functional role of the MRP1-GSTP1 relationship and that NO plays a role in macrophage cytotoxicity against tumor cells, a 2016 study assessed the role of the MRP1-GSTP1 relationship in this interaction between these cell-types [132]. This investigation demonstrated, using two well-characterized macrophage cell-types (i.e., J774 and RAW 264.7), that upon activation with LPS and interferon-γ, MRP1 expression markedly increased upon macrophage activation [132]. Moreover, the induction of iNOS and NO generation resulted in marked iron-59 release from both these cell-types, which was inhibited by *Mrp1* siRNA and the MRP1 inhibitor, MK571 [132]. Additionally, the silencing of *Mrp1* markedly increased DNIC levels in the macrophages, indicating suppression of DNIC transport out of these cells and suggesting a role for MRP1 in DNIC release [132]. Furthermore, the release of iron-59 from both macrophage cell types was significantly increased by silencing Gstp1, suggesting that GSTP1 was responsible for DNIC binding/storage [132].

Experiments in this latter study demonstrated that the expression of both GSTP1 and MRP1 were necessary to protect activated macrophages from NO cytotoxicity [132]. These results were also verified by silencing *nuclear factor-erythroid 2-related factor 2* (*Nrf2*), which transcriptionally targets MRP1 and GST P1-1 expression [132]. Silencing of *Nrf2* decreased MRP1 and GST P1-1 expression, resulting in a reduction in NO-mediated iron-59 release from macrophages and also decreased macrophage survival. Collectively, these studies indicate a mechanism of how macrophages protect themselves against NO cytotoxicity via the integration of MRP1 and GST P1-1 activity [132]. Collectively, these studies indicate that the ability of cells to transport, store, and traffic NO as DNICs between MRP1 and GST P1-1 overcomes random diffusion that is inefficient, non-targeted, and toxic.

Evolutionarily, GST enzymes have been linked to protection against NO within cells [148]. Bocedi and colleagues explored the differing affinities of 42 GST species for DNDGIC complexes and found that GSTs that have Ser and Cys residues (GSTθ, GSTβ, GSTζ, GSTε, etc.) as part of the active site did not demonstrate high enough affinity for DNGICs to sequester the complexes [148]. While the more evolved GSTs that have Tyr residues as part of their active sites (GSTα, GSTμ, and GSTπ) exhibited high affinities for the DNDGIC complexes (*K_d_* < 10^−9^ M) and were able to sequester them effectively [148]. Interestingly, these latter GSTs were also over-represented in the perinuclear region of mammalian cells [148], which further indicates that GSTs may have roles in nuclear protection against NO cytotoxicity. It was also suggested that the low affinity of the Ser and Cys-based GSTs for DNDGIC complexes may explain the sensitivity of bacteria to NO, which express this evolutionarily earlier class of GSTs [148]. 

## 9. Implications of the MRP1–GST Interaction for Understanding NO Biology: DNICs as a Common Currency for the Storage and Transport of NO

The research above highlights the significance of NO in a variety of biological processes, including vasodilation. Given the role of the MRP1 and GST P1-1 system in NO transport and storage in tumor cells (Figure 7) [132,134,162,175], it is of great interest that GST polymorphisms correlate with preeclampsia (high blood pressure in pregnancy), potentially because GSTs bind vasoactive NO as DNICs [215]. Furthermore, using a combination of congenic breeding and microarray gene expression profiling, Olson and colleagues have identified *Gst m1* as a positional and functional candidate gene involved in blood pressure regulation [216]. This latter study also identified that transgenic SHRSP rats harboring the WKY *Gst m1* gene exhibited significantly lower blood pressure and reduced oxidative stress, which are associated with hypertension and organ damage [216].

Clinical studies examining human patients have also highlighted the impact of GST M1 loss independent of traditional risk factors and have demonstrated a high correlation between GST M1 deletion and kidney and heart failure [217]. However, none of these studies have associated these functions with the MRP1-GST transport and storage mechanism for NO described herein. However, the studies linking GSTs to blood pressure support their role in regulating physiological NO metabolism, which is vital for vasoreactivity.

Critical to our hypothesis that NO is not freely diffusible, but is carefully regulated by proteins, is the concept that DNICs act as a common currency for NO storage and transport by associating with GST P1-1 and MRP1 (Figure 7). Furthermore, DNICs donate iron to tissues [218] and trans-nitrosylate targets [219,220], illustrating their bioavailability. Relative to our studies using cancer cells overexpressing GST P1-1 [162], normal cell-types express different GSTs (e.g., GST A and GST M) that can bind DNICs [184], suggesting that physiological NO metabolism could be different from that observed using the MCF7 tumor cell models [175]. Furthermore, unlike tumor cells, normal cells such as endothelial cells express eNOS, again indicating that there will be differences in NO metabolism and its interaction with iron.

As such, studies are now required to extensively assess whether the NO storage and transport mechanisms mediated by GST P1-1 and MRP1 in tumor cells are observed in normal cells. Such normal cells include hepatocytes that play crucial roles in iron metabolism and express GST A at very high levels [140], and in endothelial cells, which generate NO intracellularly, which is then released to induce smooth muscle relaxation [221,222]. The latter process is a major physiological function of NO, constituting part of the “Holy Grail” of its biological activity. In the future, this basic mechanistic dissection could lead to new therapeutics, e.g., agents mimicking DNICs to induce smooth muscle relaxation to reduce blood pressure.

The concept that DNICs function as a common currency for NO storage and transport by MRP1 and GST enzymes is significant in understanding NO metabolism. An integrated mechanism for the transport of NO in DNIC form would be invaluable to NO-producing cells. For instance, smooth muscle relaxation in blood vessels could be modulated by releasing low concentrations of DNICS from endothelial cells [223]. Identifying a physiological mechanism for NO transport and storage would overcome the random diffusion of NO, which is both inefficient and non-targeted, and is and advance on our current understanding of NO roles in iron-mediated cell signaling [179,224] and cytotoxicity [76,78,134,225].

## 10. Summary: DNIC Storage and Transport by GST and MRP1: NO and Beyond

NO has often been regarded as a toxic, freely diffusible gas. However, NO’s biological roles in macrophage cytotoxicity and vasodilation and recent developments in understanding NO regulation and storage indicate a central role for DNICs as a common currency of NO. This proposal leads to a model of NO storage and transport by MRP1 and GST P, respectively, where NO is not freely diffusible, but is rather specifically regulated and trafficked by proteins to result in a greater half-life and prevent its cytotoxicity [132,135,175]. Taking into account NO’s roles in vasodilation and many other processes, a physiological model of NO transport and storage would be valuable in understanding NO physiology and pathophysiology and could redefine NO as a tightly regulated signaling effector. Moreover, understanding these mechanisms could advance our knowledge of NO regulatory activities and could form the basis of new research that could lead to exciting new therapeutics.

## Figures and Tables

**Figure 1 molecules-26-05784-f001:**
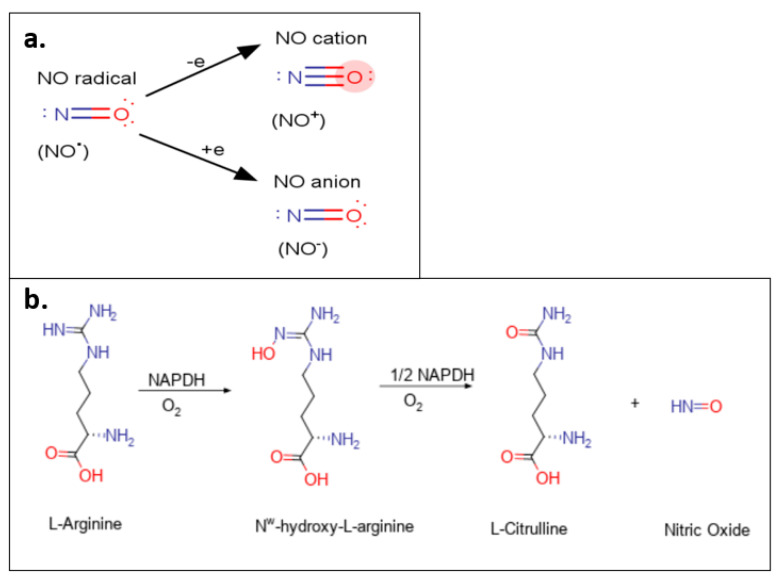
(**a**) NO redox species. NO exists as three different redox species: the NO radical (NO^•^); the nitrosonium cation (NO^+^); and the NO anion (NO^−^), which reversibly transitions between species via the addition or loss of an electron. (**b**) NO production via l-arginine. NO production occurs endogenously by the conversion of l-arginine to l-citrulline via the intermediate *N*^ω^-hydroxy-l-arginine. NADPH serves as an electron donor for both reactions.

**Figure 2 molecules-26-05784-f002:**
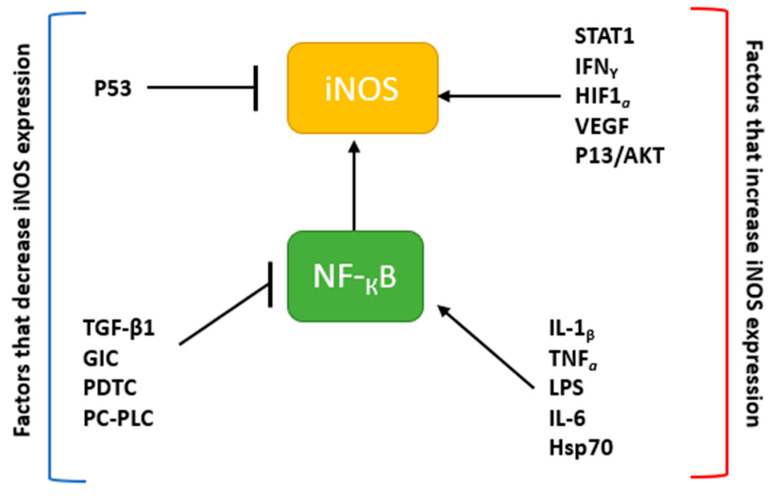
iNOS expression and related pathways. The expression of iNOS is regulated by cytokines and other factors. Factors that decrease iNOS expression include those such as the tumor suppressor, p53, as well as proteins (e.g., TGF-β) that inhibit the activity of NF-κB. On the other hand, proteins that activate iNOS such as STAT1, IFNγ, or nuclear factor-κB (NF-κB), increase the production of NO.

**Figure 3 molecules-26-05784-f003:**
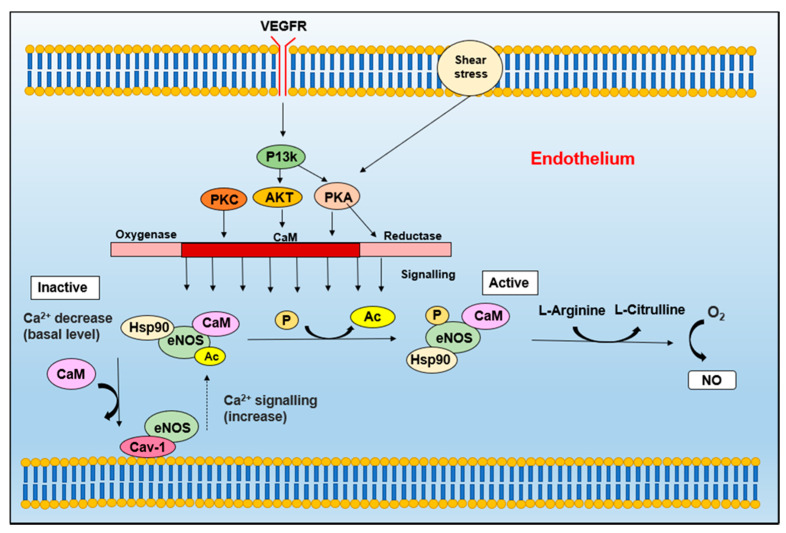
eNOS is activated by an increase in Ca^2+^ levels, which recruits CaM and Hsp90 to the enzyme, prompting caveolae (Cav-1) to dissociate from eNOS [60]. The activity of CaM and Hsp90 is promoted by EGF, which promotes CaM expression and the recruitment of Hsp90 [60]. The eNOS complex can then be activated by the phosphorylation of eNOS, which occurs via PKA (activated by shear stress), VEGF (via Ser/Thr kinase Akt), insulin, and bradykinin, which is regulated by Ca^2^^+^/calmodulin-dependent protein kinase II (CaMKII) [60]. These stimuli result in the dissociation of adenylate cyclase (Ac) from eNOS. The eNOS complex can then catalyze the production of l-citrulline and NO from l-arginine, and thus increase NO generation.

**Figure 4 molecules-26-05784-f004:**
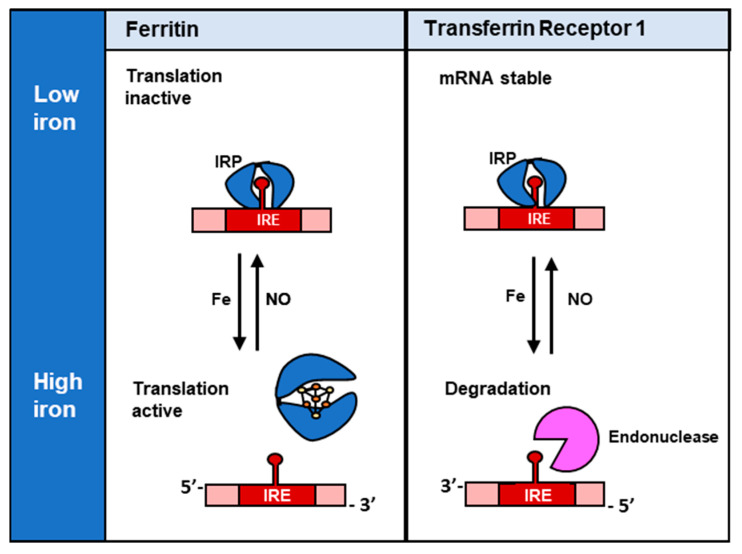
Regulation of ferritin and TfR1 by IRP1. In the absence of iron, IRP1 without an [4Fe-4S] cluster binds the 5′-IRE of *ferritin* mRNA, which sterically blocks the translation of ferritin. However, in the presence of high iron concentrations, IRP1 cannot bind to the 5′ IRE of *ferritin* mRNA due to the formation of a [4Fe-4S] cluster, and translation is active. On the other hand, when IRPs bind the 3′-IREs of *TfR1* mRNA, this interaction stabilizes the mRNA and protects it from endonucleases so that TfR1 translation can occur. Whereas, under high iron concentrations, IRPs cannot bind to the 3′-IREs of the *TfR1*, and the mRNA is exposed and degraded by endonucleases.

**Figure 5 molecules-26-05784-f005:**
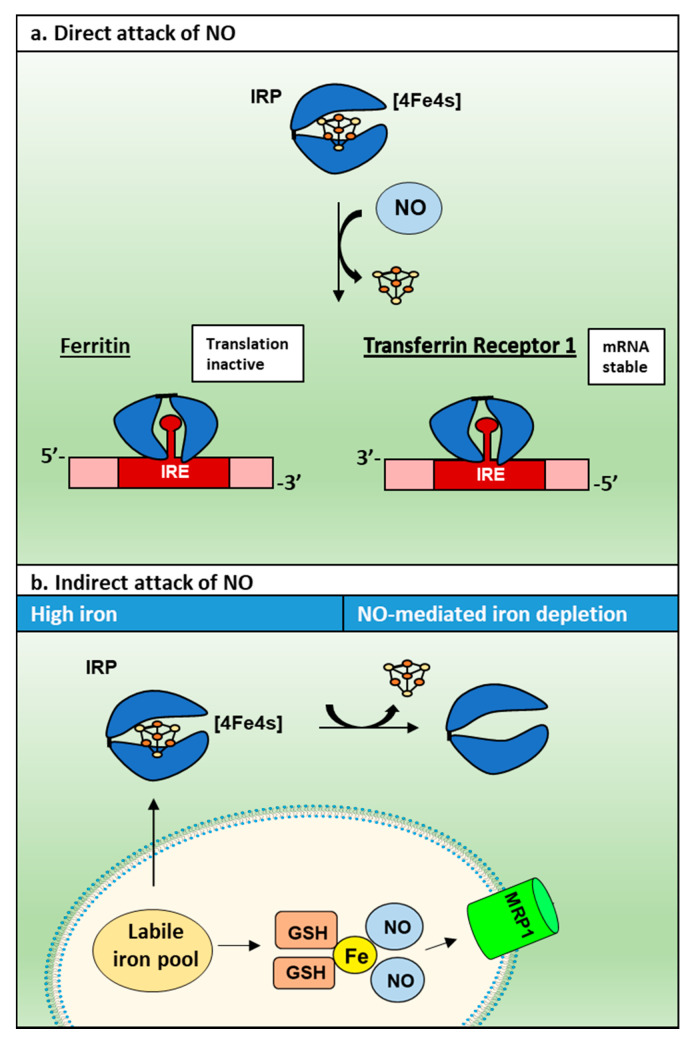
The direct and indirect impact of NO on IRP1 and IRE-binding. In the presence of NO, two mechanisms result in the loss of the [4Fe-4S] cluster: (**a**) direct attack of NO inducing disassembly of the cluster; and (**b**) an indirect mechanism where NO binds intracellular iron to induce its release from cells, resulting in iron depletion that therefore prevents [4Fe-4S] cluster biogenesis.

**Figure 6 molecules-26-05784-f006:**
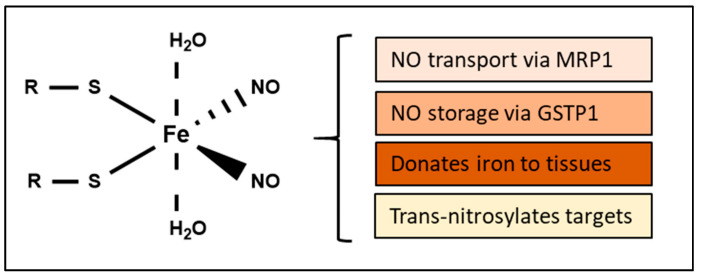
The structure and biological functions of DNICs. R may be either glutathione or cysteine.

**Figure 7 molecules-26-05784-f007:**
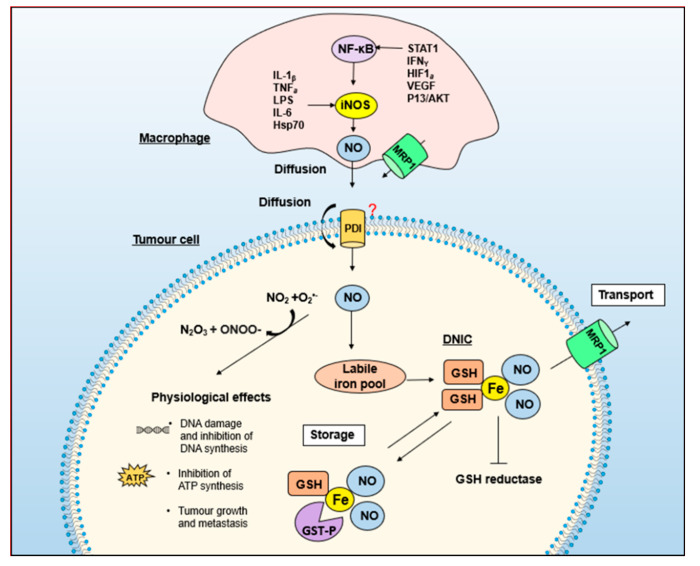
Proposed mechanisms involved in inhibiting tumor cell metabolism after co-cultivation of tumor cells with NO-generating activated macrophages. Cytokines and other agents (LPS, IFNγ, STAT1, and TNF-α, etc.) promote the synthesis of iNOS-derived NO in macrophages [16]. NO can then diffuse from the cell or form a DNIC complex and be actively transported out of the cell [132,135]. NO is transported into the target tumor cell via diffusion or actively by protein disulfide isomerase (PDI) through trans-nitrosylation [177]. NO within cells has several physiological effects [178,179] and can react with iron and GSH to create DNICs. MRP1 functions to export DNICS out of the cell to facilitate iron and GSH efflux, which results in iron depletion and inhibition of tumor cell proliferation.

**Figure 8 molecules-26-05784-f008:**
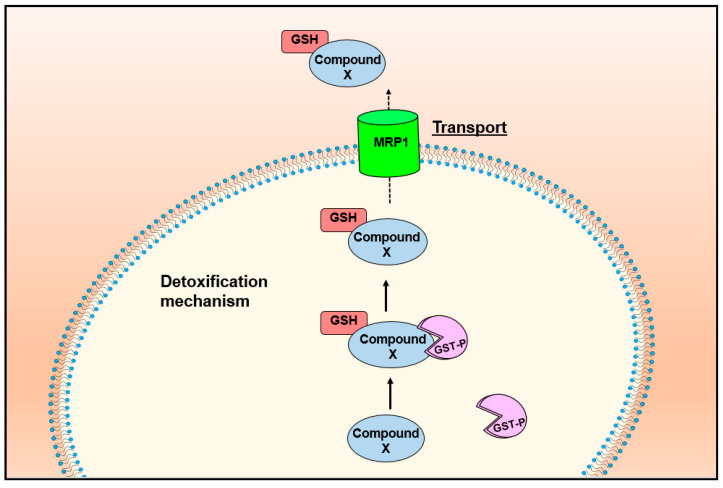
Detoxification mechanism between MRP1 and GST enzymes. GST enzymes conjugate GSH to toxic compounds to target them for export out of the cell by the GSH-conjugate transporter, MRP1.

## Data Availability

Not applicable.

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
