# Peer review of "The Relationship of Glutathione-S-Transferase and Multi-Drug Resistance-Related Protein 1 in Nitric Oxide (NO) Transport and Storage"

_molecules, 2021, doi:10.3390/molecules26195784_

Round 1

Reviewer 1 Report

This work presents a nice review of the transport system of nitric oxide in cells.

Fig. 1 should be improved to show in the structural modell the radical character of NO and the charge of nitrosonium cation and nitroxyl anion.

In the abstract no abbreviations should be used.

Reviewer 2 Report

In this manuscript, the authors focused on the biological functions, transport, and the storage of nitric oxide. The concept that NO is stored as the DNICs complex is of great interest. The manuscript is well written, which can be published after considering the following comments.

The authors have mainly focused on mammalian cells, does this model also work for bacterial cells?

Apart from the nitric oxide synthases, other enzymes like xanthine oxidoreductase could also mediate NO formation, and of importance, the [2Fe–2S] clusters are required for the synthesis of this enzyme. Will the authors think the binding of NO to iron (heme and iron-sulfur cluster) could lead to a feedback on the NO generation? How would the author consider the influences of DNICs on the other cellular metal ion pools, for instance, copper and molybdenum, which have close links with iron metabolism.

Reviewer 3 Report

This review features discussions on the physiological function and the metabolism of Nitric Oxide (NO) and dinitrosyl-dithiol-iron complex (DNICs). Moreover, the authors summarized the relationship of glutathione-S transferase (GST) and multi-drug resistance-related protein 1 (MRP1) in DNICs transport and storage. The authors claim that a physiological model of NO transport and storage would be valuable in understanding NO physiology and pathophysiology. This review contains valuable information on the role of GST and MRP1 in NO transport and storage. They summarized the important findings on the role of NO, which makes it an interesting topic. The following points should be clarified.

  1. In Figure 2, the intention of the authors is difficult to understand. The authors should make the concept expressed in Figure 2 more clearly, especially the relationship between the left and right arrows and the pathways in the middle.

  1. The authors focused on DNICs in this review. Adding an illustration of DNICs metabolism and the structural formula would help us to understand this review.

  1. What does “GIC” stand for in Figure 2?
